# Impact of COVID-19 lockdown on PM concentrations in an Italian Northern City: A year-by-year assessment

Daniele Pala[1]*, Vittorio Casella[2], Cristiana Larizza[1], Alberto Malovini[3], Riccardo Bellazzi[1]

**1** Department of Electrical, Computer and Biomedical Engineering, University of Pavia, Pavia, Italy, **2** Department of Civil Engineering and Architecture, University of Pavia, Pavia, Italy, **3** Istituti Clinici Scientifici Maugeri SpA SB IRCCS, Pavia, Italy

* daniele.pala@unipv.it

**Data Availability Statement:** Data are uploaded to the Open Science Framework repository, the DOI is 10.17605/OSF.IO/4UZPA."

## Abstract

In the last century, the increase in traffic, human activities and industrial production have led to a diffuse presence of air pollution, which causes an increase of risk of several health conditions such as respiratory diseases. In Europe, air pollution is a serious concern that affects several areas, one of the worst ones being northern Italy, and in particular the Po Valley, an area characterized by low air quality due to a combination of high population density, industrial activity, geographical factors and weather conditions. Public health authorities and local administrations are aware of this problem, and periodically intervene with temporary traffic limitations and other regulations, often insufficient to solve the problem. In February 2020, this area was the first in Europe to be severely hit by the SARS-CoV-2 virus causing the COVID-19 disease, to which the Italian government reacted with the establishment of a drastic lockdown. This situation created the condition to study how significant is the impact of car traffic and industrial activity on the pollution in the area, as these factors were strongly reduced during the lockdown. Differently from some areas in the world, a drastic decrease in pollution measured in terms of particulate matter (PM) was not observed in the Po Valley during the lockdown, suggesting that several external factors can play a role in determining the severity of pollution. In this study, we report the case study of the city of Pavia, where data coming from 23 air quality sensors were analyzed to compare the levels measured during the lockdown with the ones coming from the same period in 2019. Our results show that, on a global scale, there was a statistically significant reduction in terms of PM levels taking into account meteorological variables that can influence pollution such as wind, temperature, humidity, rain and solar radiation. Differences can be noticed analyzing daily pollution trends too, as—compared to the study period in 2019—during the study period in 2020 pollution was higher in the morning and lower in the remaining hours.

**Funding:** The work has been funded by the European Commission with Grant Agreements GA-727816 and GA-101016233. All authors received funding. The funders had no role in study design, data collection and analysis, decision to publish, or preparation of the manuscript.

## Introduction

In the last decades, air pollution has become a major threat to health and wellbeing in several countries of the world, most of them being urban or highly populated areas. Scientific investigations showed that exposure to air pollution can lead to an increased risk of developing or exacerbating several diseases, in particular respiratory and cardiovascular ones [1]. The prevalence of diseases such as asthma, type 2 diabetes and cardiovascular disorders is also increasing in most of the world [2–4]. While public health organizations, together with non-profit associations and governments are struggling to apply "green solutions" to contain the increase in pollution levels, a sudden reduction of air pollutants concentration was observed in many countries between the end of the year 2019 and the year 2020. During this period, most of the productive activities throughout the world had to come to an unexpected stop due to the pandemic caused by the new SARS-CoV-2 virus, and the related diseases, COVID-19. This virus was firstly reported in Wuhan, China, and quickly spread causing a high number of intense flu-like syndromes and cases of atypical pneumonia. Compared to influenza viruses, this unknown pathogen showed abnormally high contagious strength, with higher hospitalization and death risks [5]. Despite the initial efforts to contain the disease, the virus spread in several Asian countries outside of China, and at the end of February 2020 the first European case unrelated to the Asian outbreak was found in northern Italy, in the city of Codogno in Lombardy. In the following weeks, Italy suffered a tremendous increase in terms of COVID-19 cases, forcing the government to take drastic actions. Thus, Italy was the first western country to apply severe measures such as a general lockdown, with most of the population confined at home and a shutdown of all nonessential productive activities and services.

The first area where schools and factories have been closed included the most densely populated and industrialized regions of northern Italy, located in a large area named Po Valley. Among other aspects, this area is known to be among the most polluted ones in Europe [6], since several air pollutants, including particulate matter, often rise to dangerous levels. This phenomenon is due to an unfortunate combination of factors such as high population density, intense industrial activity and geographic position, as the valley is surrounded on three sides by mountains. Starting from China in December 2019, a reduction of human related pollution was observed in all areas where the lockdown was applied by governments. In several parts of the world air pollution dropped to the lowest levels in decades [7, 8]; this happened particularly in Asian countries such as China and India. In Italy, although a similar effect has been observed, the reduction in terms of concentration was not evident for all pollutants. A significant decrease in nitrogen dioxide and sulfur dioxide was observed during the first period, whereas no drop in particulate matter ($PM_{10}$ and $PM_{2.5}$) was recorded [9]. Peculiar meteorological conditions such as high concentration of sea salt and Saharan sand brought by Eastern winds at the beginning of the lockdown and a coincidental increase in the wind speed for several days could have contributed to this phenomenon [10]. Meteorological factors are known to play a considerable role in modulating particulate matter and other pollutants concentration: wind tends to disperse them, while lower or higher temperatures can create favorable conditions for higher concentrations. The presence of confounders makes it difficult to assess the real impact of traffic, industrial activities and house heating on the pollution levels, especially because they can potentially interact in complex ways. The sudden lockdown condition in which Italy entered between February and March 2020 allowed to analyze how vehicular traffic and factories activities impact on air pollution during the daily life. Although traffic was limited to essential transportations and factories not producing essential goods were closed during this period, results from studies were controversial due to the extreme variability in terms of weather condition typically characterizing the month of March [11]. Air pollution is a

typical problem in the Po Valley during winter months [12], when cold dense air tends to stagnate in the lower layers of the atmosphere. The phenomenon is less common in the warm season as air is less dense and local breezes are more frequent.

In this paper we present the results from a study performed in Pavia, located in the Po valley in Italy, where 45 air quality sensors have been deployed in the urban area in the context of the European project PULSE (Participatory Urban Living for Sustainable Environments [https://www.project-pulse.eu/]), allowing to create a dense network of pollutants measurements.

The primary objective of the analysis was to assess the impact of the lockdown on the urban PM pollution. To this aim we analyzed $PM_{2.5}$ and $PM_{10}$ data coming from the available sensors with high spatial and temporal resolution and compared measurements performed from the end of February to the beginning of April of 2020 with those deriving from the same period in 2019. Analyses consisted of several steps: we first analyzed data from two official monitoring stations in Pavia, then we investigated the effect of meteorological conditions (wind and temperature) on the measurements, and finally we estimated the adjusted mean variations in terms of $PM_{2.5}$ and $PM_{10}$ between the study periods in years 2019 and 2020 at a whole data level and at a single sensor level stratified by daily hours.

## Materials and methods

### The PULSE project

New public health initiatives are emerging to face the new challenges brought by the increase of population in the urban areas and the global changes in air pollution and lifestyle. Among these, the PULSE project was funded by the European Commission to perform heath research in big cities across Europe, Asia and the U.S.A. through a collaboration between universities, research centers and city councils. The project started during the fall of 2016 and lasted until April 2020 with clinical focus on the link between air quality and asthma, physical inactivity and type 2 diabetes and on the influence of exposures on health and wellbeing in general. A complex multi technological platform was created in the context of the PULSE project and tested in seven cities: Barcelona (Spain), Birmingham (United Kingdom), Keelung (Taiwan), New York (U.S.A.), Paris (France), Pavia (Italy) and Singapore. This platform is based on a sophisticated data exchange infrastructure that involves data coming from different sources, including satellite data, open data and dedicated sensing technology for air quality.

**The sensors network in Pavia.** The city of Pavia, Italy, was one of the test sites of the PULSE system. Although Pavia is a rather small city (about 72,000 inhabitants), its university and its particular position in the geographical area of the Po Valley makes it a suitable location to carry on research and innovation projects to study air quality and personal exposure to pollutants, being air pollution and its related health risk a frequent problem that affects this area during winter months.

One of the key innovation aspects of PULSE is the use of data with high spatial and temporal resolution. The increase in terms of resolution can be beneficial to study phenomena of public health interest at a neighborhood level, thus taking into account social and health disparities that often characterize large urban environments, as highlighted by studies we performed in the context of the project [13, 14]. Despite this necessity, urban data is rarely collected with a sufficient spatial granularity, due to the high costs and difficulties of the process. Taking air quality as an example, pollution is usually measured by high quality monitoring stations deployed in the cities by official agencies or state departments. Being these instruments expensive and requiring constant maintenance they are usually deployed in small numbers. As an example, the entire city of New York has only 13 monitoring stations while Pavia has 2 stations over an area of about 65 km$^2$. For this reason, low-cost and portable

sensing technologies are becoming common [15] but generating controversial results: while the frequency and easiness of measurements increase with low-cost technology, the quality of the data collected is often limited.

A total number of 45 low-cost PurpleAir sensors [16] were purchased and installed throughout Pavia in agreement with the municipality and with the collaboration of a few private citizens within the PULSE project. These sensors are small outdoor devices that measure three types of pollutants, i.e. $PM_1$, $PM_{2.5}$ and $PM_{10}$, and temperature and humidity as meteorological data. These instruments are easy to use as they can be installed on private apartments' balconies and require only a Wi-Fi connection and an electrical outlet. The unit price is relatively low (about $250), as they measure particulate matter using a simple laser counter, that has low production costs and requires low maintenance, although more prone to anomalous readings than more expensive technologies. The network created by these sensors was added to already existing official monitoring stations, property of *ARPA Lombardia* (Environmental Protection Regional Agency of the region of Lombardy), a public agency that aims at measuring environmental data in the Lombardy region in Italy—where Pavia is located. Through numerous sensors scattered throughout the region, ARPA collects a large quantity of data about air quality, meteorology, agriculture, sole status, etc.

In Pavia, there are two official ARPA air quality monitoring stations, they are high-quality fixed stations that measure several pollutants ($NO_X$, $SO_2$, $CO$, $O_3$, $PM_{10}$ and $PM_{2.5}$) at regular time intervals. These sensors are calibrated continuously using commercially available instruments that can be used as reference according to Italian or European laws [17] and therefore can be considered very reliable. Weather parameters are collected as well, and data is freely available upon request on a dedicated portal. An analysis has been performed to evaluate whether measurements performed by the Purple Air sensors were different from the ones performed by one of the official monitoring stations belonging to the local environmental agency ARPA. It was observed that, despite a small offset, the Purple Air sensors' measurements were highly correlated with the official ARPA ones (correlation > 0.9). Thanks to the large number of available devices, a dense sensors network was created, allowing to derive high-quality homogeneous maps through interpolation and to use them in the monitoring of pollution levels and personalized risk for the citizens that move around the city. Besides defining a spatially dense network, these sensors allow also for high temporal granularity in the measurements, by quantifying $PM_1$, $PM_{2.5}$, $PM_{10}$ every 80 seconds. The geographical position of the Purple Air sensors located in Pavia can be visualized on the real time map available on the Purple Air website, showing the latest sensors' measurements [18].

## Data analyzed

Two main data sources have been used in this study: air quality measurements coming from the Purple Air sensors and weather data (wind speed, temperature, humidity, precipitations and solar radiation) acquired from the ARPA official database. In particular, used sensors measure all kinds of particulate matter ($PM_1$, $PM_{2.5}$ and $PM_{10}$), but only $PM_{2.5}$ and $PM_{10}$ have been considered in our analysis, since they are well-agreed indexes of air pollution. Even though the Purple Air sensors themselves measure weather data (temperature and humidity), these data were not of interest in the PULSE project and therefore the accuracy of weather measurements was not determined. For this reason, we decided to use the official ARPA sensors to measure meteorological variables included in the study. Being Pavia a small city, it is not expected to observe significant meteorological differences across different areas of the city.

We considered hourly pollution measurements data during two periods of time: from February 24th, 2020 (00:00 CET) to April 2nd, 2020 (24:00, CEST) and the same period during year

2019. During 2020 the time period considered started in correspondence of the days after the identification of the first COVID-19 cases in northern Italy, with the consequent closing of schools and of most of the commercial activities and the deriving reduction in terms of urban mobility (the so-called "lockdown"). The same temporal period during year 2019 has been considered as reference, to estimate $PM_{2.5}$ and $PM_{10}$ variations between the two years. Meteorological data (measurements of hourly average wind speed, maximum wind gusts, air temperature, humidity, precipitations and solar radiation) were collected with the same hourly temporal granularity as PM. Only sensors with available measures for both 2019 and 2020 were included in the analysis.

**Data calibration.**   Data measured by Purple Air sensors were calibrated using the official ARPA monitoring stations measurements, using data of a single sensor located close to one of the ARPA stations (PurpleAir sensor 3, PA-S3). Linear regression was used to calibrate the sensor data according to the following model:

$$y = m \cdot x + b$$

Where $y$ indicates data coming from the Purple Air sensor, $x$ those coming from the ARPA monitoring station, and $m$ and $b$ are the regression coefficients, representing the slope and the intercept respectively.

Using measurements with the same temporal granularity (measurements were aggregated by day as the ARPA data was available only with this temporal granularity), the values of $m$ and $b$ were estimated for $PM_{2.5}$ and $PM_{10}$ independently. Data from the considered sensor and from the ARPA station showed evidence of strong correlation (r = 0.83 and r = 0.80 for $PM_{2.5}$ and $PM_{10}$ respectively). Once the parameters were estimated, all sensors' measurements were corrected by the inverse formula:

$$\hat{x} = \frac{(y - b)}{m}$$

Specifically, by comparing the $PM_{2.5}$ measurements between the Purple Air and the ARPA sensors we obtained $m$ = 1.5755 and $b$ = -5.7709, while $m$ = 1.1851 and $b$ = -16.1240 when comparing $PM_{10}$ values. Thus, by defining $PM_{2.5}$' and $PM_{10}$' as the crude values measured by our sensors, the corresponding scaled PM2.5 and PM10 values have been calculated as follows:

- $PM_{2.5}$ = ($PM_{2.5}$' + 5.7709) / 1.5755

- $PM_{10}$ = ($PM_{10}$' + 16.1240) / 1.1851

Scaled $PM_{2.5}$ and $PM_{10}$ measurements were then used for the analyses reported in the next sections.

Considering that the Purple Air sensors have all the same hardware, we co-located one sensor close to the ARPA one to estimate the correction to be made, and then apply it to all the other Purple Air sensors, rather than co-locate all sensors close to the ARPA ones. Furthermore, an internal study performed at the University of Pavia locating seven Purple Air sensors in the same place showed evidence of strong correlation between measures performed by co-located sensors. Results are reported in S1–S4 Figs, showing the correlation matrices and the range of measured values for each sensor on a selected day (July 24[th] 2019). Considering that Pavia is a small city with no significant climatic differences across different zones, we assumed that possible changes in performance over time should not affect some sensors more than others.

## Data pre-processing

Only data from 24 out of the 45 Purple Air sensors available in Pavia were analyzed since some of these were not active during both years 2019 and 2020. Scatterplots and boxplots by sensor were generated to visually inspect the correlation between $PM_{2.5}$ and $PM_{10}$ concentration ($\mu g/m^3$) and to identify potential outlier measurements. Visual inspection of boxplots allowed identifying and removing 4 outlier values corresponding to $PM_{10}$ and $PM_{2.5} > 250$, for a total number of 39,926 measurements passing quality control.

## General pollution trend over the years

Before comparing 2019 and 2020 data by statistical methods, it is useful to check for the presence of potential increasing or reducing trends in PM levels in the study area. The continuous technological improvements in vehicular traffic and energetic efficiency are presumably leading to a slow decrease of average pollution levels in Europe, and this could be a confounding element in the evaluation of the lockdown effects using data from the previous year, since an observed reduction could be due to the general trend instead of the lockdown itself. According to data in literature and past articles, there is some evidence of a $PM_{10}$ and $PM_{2.5}$ reduction trend in all the area in the last years [19] although with notable fluctuations. Looking at an article published by the Italian National Environmental Protection System [20], it can be noticed that the reduction trend appears less evident after 2018, with even a little increase, probably not significant, in the $PM_{10}$ concentrations in 2020. The article itself states that the meteorological variability could have played an important role in the measurements' variations, as in 2019 and 2020 temperatures were generally higher and precipitations lower than the previous years. Therefore, we assumed that the general trend characterizing the last years probably did not significantly influence the difference in PM concentrations between 2019 and 2020.

## Statistical and data mining methods

The Wilcoxon Mann Whitney test was applied to compare numeric variables distribution between groups. Multivariate regression trees were fitted to identify informative wind speed cut off values able to discriminate subpopulations of $PM_{2.5}$ and $PM_{10}$ measurement values. The one sample t-test was applied to test the null hypothesis that the mean variation in terms of PM pollutants between years was zero. The Spearman correlation coefficient *r* and corresponding 95% confidence interval (95% CI) were computed to estimate the correlation between numeric variables by a bootstrap approach based on 1,000 replicates. Linear mixed model regression was then used to estimate the adjusted mean variation in terms of $PM_{2.5}$ and $PM_{10}$ between the study periods in years 2019 and 2020, using data matched between years by measurement month, day and hour. Year (binary: 2019 coded "0", 2020 coded "1"), working day (binary: week end coded "0", working day coded "1"), wind (numeric continuous, expressed as m/s), temperature (numeric continuous, expressed as ˚C), humidity (numeric continuous, expressed as %), precipitations (numeric continuous, expressed as mm), solar radiation (numeric continuous, expressed as $W/m^2$) and sensors elevation upon the sea level (continuous, expressed as meters) were included as fixed effect terms, while a single variable resuming the sensors ID/month/day/hour of measurement was used as random effect grouping variable. The interaction between year and daily hour was also included in order to estimate the adjusted mean variation in terms of pollutants concentration by daily hours interval. Analyses were performed at a whole data level and then repeated by sensor.

An alternative approach consisting of a modification of the method proposed by Venter et al. [21] has been also applied to estimate the potential impact of the lockdown on pollutants concentration during year 2020. The method is described in the detail in the S1 File section.

Analyses were performed using the R statistical software tool version 3.6.1 (www.r-project.org). Regression trees were fitted by the *rpart* fuction implemented in the *rpart* package. Spearman coefficients and corresponding 95% CI were computed by the *spearman.ci* function implemented in the R package called *RVAideMemoire*. Linear mixed model analysis was performed by the *lmer* function implemented in the R package called *lme4*. MATLAB (R2020b Version) was used for sensors' calibration, analyses regarding the official ARPA data and the creation of visual representations of the PMs trends in the two years.

## Results

### Analysis of the official ARPA data

ARPA owns two sensors in the urban area of Pavia: the first one (ARPA-S1) is located in a square very close to the city center and along one of the most important traffic arteries of the city, characterized by high traffic during the rush hours. The second one (ARPA-S2) is located in a residential area, densely populated but less prone to traffic. Unfortunately, ARPA historical data are available with a daily temporal granularity for $PM_{10}$ only. As a matter of fact, ARPA-S1 open data does not contain $PM_{2.5}$ measurements and ARPA-S2 open data are characterized by an extremely high missing data fraction in the considered period.

When comparing measurements distribution between years it was possible to observe different behaviors of the two sensors, as ARPA-S1 showed a significant decrease in the $PM_{10}$ levels during the year 2020 (median decreasing from 42 μg/m$^3$ to 26 μg/m$^3$, p-value = 0.0016), whereas ARPA-S2 measurements distribution did not show significant variations (p-value = 0.4571). The difference between the two sensors is visible also in Fig 1, where $PM_{10}$ measurements of both sensors in the study periods in 2019 and 2020 are shown. This analysis was performed on daily averaged data for each sensor, the only data available from the ARPA dedicated portal.

These differences are presumably due to the differences characterizing the neighborhoods in which the two sensors are located. This enlightens the necessity to take local factors into account when analyzing punctual air quality data coming from air quality sensors, as measurements can be highly influenced by the environmental and meteorological contest in which the sensor is located.

Besides $PM_{2.5}$ and $PM_{10}$, when focusing on other kinds of pollutants measured by ARPA sensors (CO, $NO_2$, $SO_2$ and NO) it has been possible to observe a statistically significant

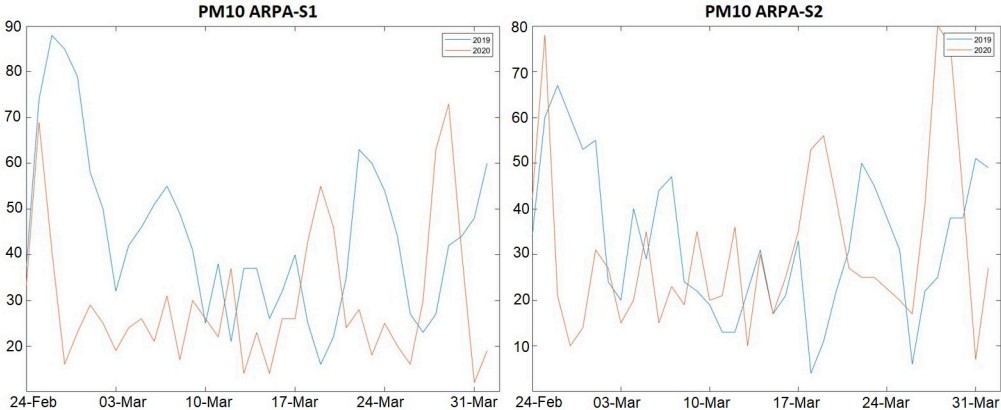

**Fig 1. $PM_{10}$ daily trends during 2019 and 2020 study periods measured by the two ARPA monitoring stations.**

**Table 1. Results of the Wilcoxon Mann Whitney test on ARPA data.**

| Pollutant | Median 2019 (Q1:Q3) | Median 2020 (Q1:Q3) | p-value |
|---|---|---|---|
| CO | 0.6 (0.6:0.7) | 0.7 (0.5:0.8) | 0.1295 |
| $NO_2$ | 34.0 (20.6:47.9) | 20.9 (13.3:31.7) | <0.0001 |
| $SO_2$ | 5.2 (4.3:6.1) | 1.8 (1.4:2.3) | <0.0001 |
| NO | 42.7 (27.7:64.5) | 28.2 (18.0:42.5) | <0.0001 |

The table reports the median values of the pollutants in the considered periods in 2019 and 2020, together with their 25th and 75th percentiles, and the tests' p-value. Apart from CO, all pollutants had a significant decrease in 2020 compared to 2019, even neglecting the effect of meteorological variables.

reduction in terms of all pollutants except for CO during the study period in 2020 compared to the same period in 2019 (p < 0.0001) (Table 1).

## Effect of potential confounders on pollutants concentration

After outliers removal a total number of 39,926 measurements passed quality control. An exploratory analysis has been performed to inspect how the pollution trends have changed during the different time periods and how weather conditions influenced this change. Looking at the absolute quantities reported in Fig 2, showing scaled data from the sensor PA-S3 (the same used for data calibration), a very irregular trend in pollution concentration can be observed during both years, suggesting that external factors can influence these measurements.

Looking at the two plots, it is possible to observe that high wind speed peaks correspond generally to lower concentrations of $PM_{10}$. The same trend has been observed for $PM_{2.5}$ (S5 Fig).

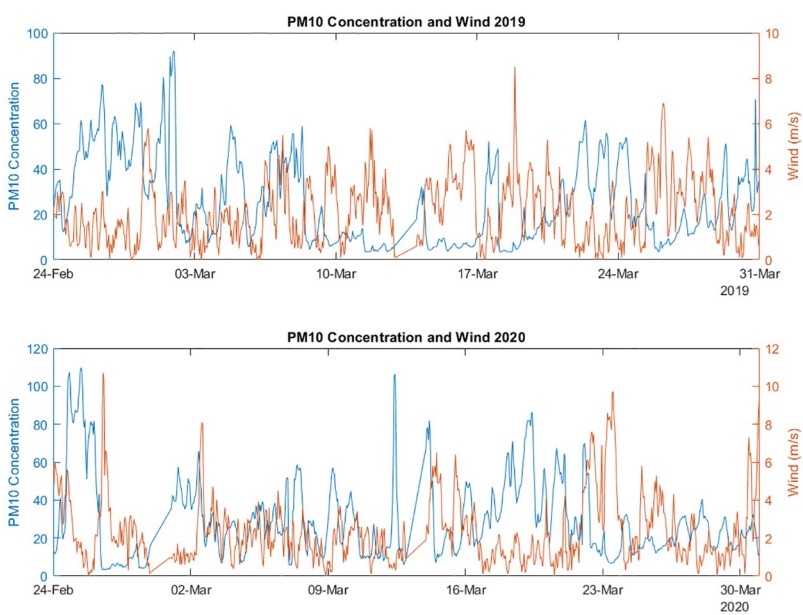

**Fig 2. $PM_{10}$ concentration and wind speed in the two considered periods in 2019 (upper plot) and 2020 (lower plot).**

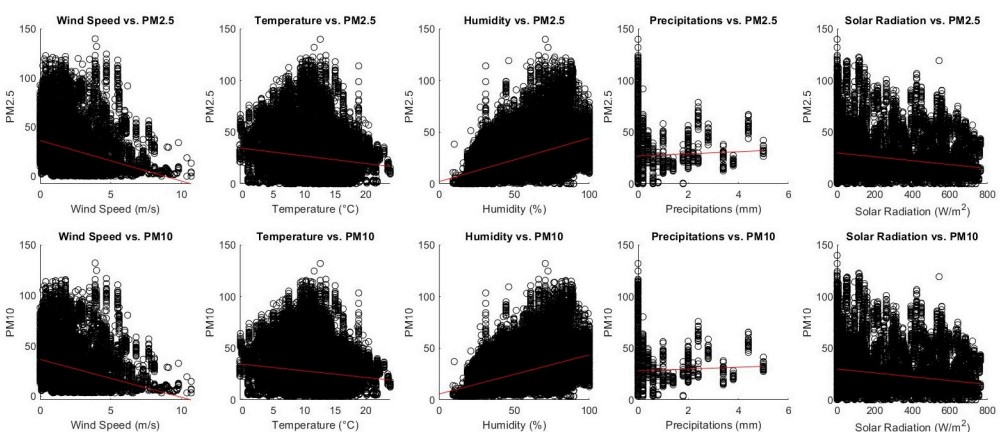

**Fig 3. Scatterplots describing the correlation between wind speed, temperature, humidity, precipitations, solar radiation and PM concentration.**

Similarly, the effect of temperature on PM concentration (scaled values) has been inspected. Plots in S6 and S7 Figs show an apparently milder relationship between temperature and pollutants concentration.

The relationship between wind speed (m/s), temperature (˚C), humidity (%), precipitations (mm), solar radiation (W/m$^2$) and pollutants' concentration using scaled data pooled from both 2019 and 2020 study periods was assessed by visual inspection of the scatterplots reported in Fig 3.

Plots in Fig 3A and 3F confirm a mild negative correlation between wind speed and both $PM_{2.5}$ and $PM_{10}$ levels ($PM_{2.5}$: r = -0.40, 95% CI = -0.41:-0.39; $PM_{10}$: r = -0.40, 95% CI = -0.41:-0.39), especially for high wind speed values. High wind speed values could reduce pollutants concentration, representing a potential confounder when comparing PM levels between 2019 and 2020. Further, $PM_{2.5}$ and $PM_{10}$ levels correlated positively with humidity ($PM_{2.5}$: r = 0.47, 95% CI = 0.46:0.48; $PM_{10}$: r = 0.47, 95% CI = 0.46:0.47, Fig 3C and 3H).

Weaker and negative correlation was observed between temperature and pollutants concentration ($PM_{2.5}$: r = -0.21, 95% CI = -0.22:-0.20; $PM_{10}$: r = -0.20, 95% CI = -0.21:-0.19, Fig 3B and 3G) and between pollutants concentration and solar radiation ($PM_{2.5}$: r = -0.19, 95% CI = -0.20:-0.18; $PM_{10}$: r = -0.19, 95% CI = -0.20:-0.18, Fig 3E and 3J).

The evidence of correlation between precipitations and pollutants concentration was almost null (PM2.5: r = 0.02, 95% CI = -0.01:0.03; $PM_{10}$: r = 0.02, 95% CI = 0.01:0.03, Fig 3D and 3I).

Based on the scatterplots in Fig 3 it was possible to observe a nonlinear relationship between wind speed and pollutants concentration (Fig 3A and 3F). In order to identify informative wind speed cut off values able to distinguish subpopulations of measurements, univariate regression trees were fitted including wind speed as predictor while $PM_{2.5}$ and $PM_{10}$ as dependent variables in turn.

By visual inspection of the cross-validation results of the unpruned trees it was possible to observe that the first split caused a major reduction of the relative cross validation error of about 10% for both pollutants, further splits induced less relevant reductions (~ 2%) as shown in S8 Fig. Thus, by imposing a single split to the regression tree algorithm, a wind speed of 2.15 m/s was identified as the most informative threshold to stratify both $PM_{2.5}$ and $PM_{10}$ levels. Pollutants concentration measured when the wind speed was ≥ 2.15 m/s were significantly lower compared to those performed when the wind speed was below the threshold (p-value < 0.0001).

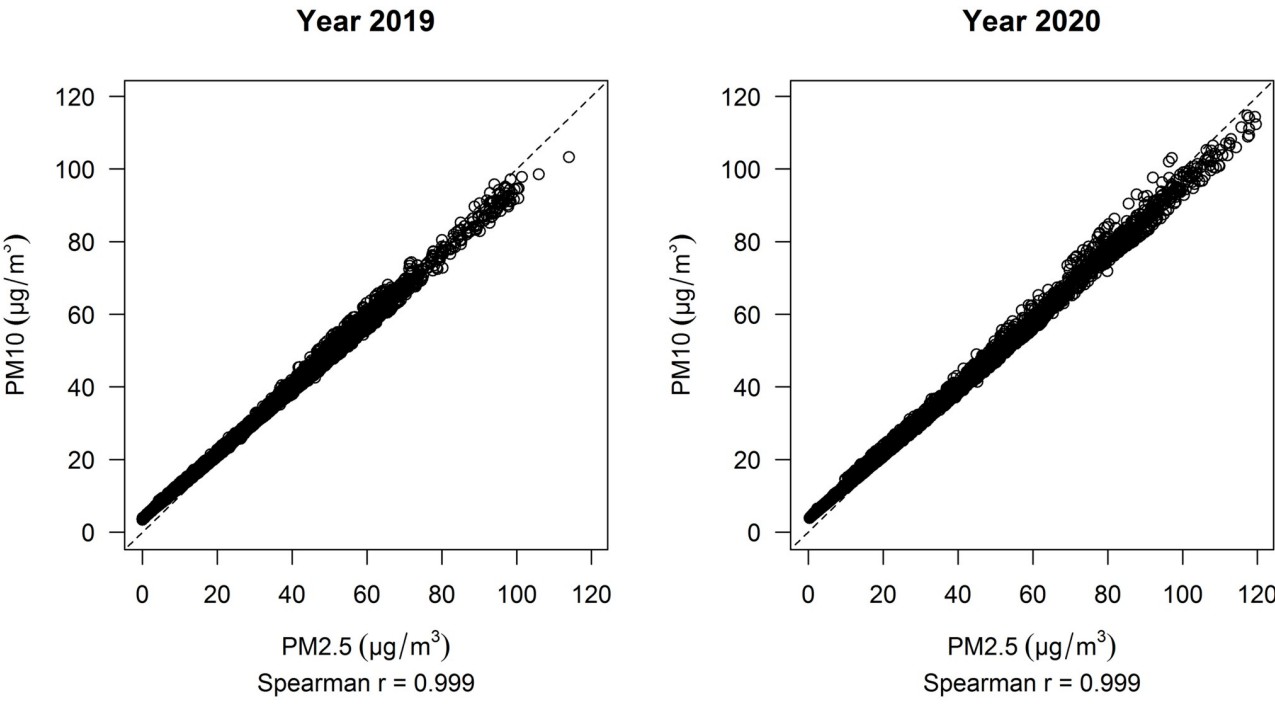

**Fig 4. Correlation between PM2.5 and PM10 during years 2019 and 2020.**

Plots in Fig 3C and 3H evidenced that pollutants concentration did not vary with respect to humidity when humidity values were below ~ 20%, highlighting a potential bias in terms of measurements accuracy when the confounder value is below this threshold.

It was then decided to focus on a subset of 22,240 measures performed when the wind speed was below 2.15 m/s and humidity > 20% to avoid confounding effects.

## Matching measurements between years

PM measures were then matched between 2019 and 2020 by sensor, month, day and hour to further reduce the potential impact of confounders when assessing their variation between years. Furthermore, all data from a single sensor (PA-S24) were also removed due to the low number of measurements available compared to the others (n = 154 vs. n ≥ 436). A total number of 5,452 paired measures (year 2019: 5,452 measures, year 2020: 5,452 measures) from 23 sensors were included in the analyses on the basis of the matching criteria. S1 Table reports the number of measurements available by sensor.

The correlation between $PM_{2.5}$ and $PM_{10}$ measures during the considered periods in 2019 and 2020 as well as the correlation between variations in terms of $PM_{2.5}$ and $PM_{10}$ between 2019 and 2020 using matched data was extremely high (Spearman r > 0.99), as shown in Fig 4 and S9 Fig.

**Variation in terms of pollutants concentration between 2019 and 2020.** The variation in terms of $PM_{2.5}$ and $PM_{10}$ concentration between the study periods in 2019 and 2020 has then been estimated, results are reported in Table 2 and show no statistically significant variations in terms of average PM2.5 (p-value = 0.177) or PM10 (p-value = 0.230).

Two multivariate linear mixed effects regression models have been fitted to quantify the variation in terms of $PM_{2.5}$ and $PM_{10}$ between the study periods in 2019 and 2020, adjusting

**Table 2. PM2.5 and PM10 distribution during 2019 and 2020.**

| | | PM2.5 | | |
|---|---|---|---|---|
| Year | N | Mean ± SD | Median (Q1:Q3) | p |
| 2019 | 5,452 | 34.2 ± 21.02 | 34.6 (15.63: 50.79) | |
| 2020 | 5,452 | 33.68 ± 23.69 | 28.11 (17.55: 45.12) | |
| Variation | 5,452 | -0.52 ± 28.56 | -0.38 (-17.92: 17.33) | 0.177 |
| | | PM10 | | |
| Year | N | Mean ± SD | Median (Q1:Q3) | P |
| 2019 | 5,452 | 34.62 ± 19.18 | 34.76 (17.69: 49.74) | |
| 2020 | 5,452 | 34.2 ± 21.76 | 28.87 (19.5: 44.18) | |
| Variation | 5,452 | -0.43 ± 26.19 | -0.14 (-16.12: 15.73) | 0.230 |

Year = analyzed year or variation between years; Mean ± SD = mean value and standard deviation of pollutants concentration during years 2019, 2020 and variations between 2019 and 2020; Median (Q1:Q3) = median value and 25th: 75th percentiles of pollutants concentration during years 2019, 2020 and variations between 2019 and 2020; p = p—value from the paired samples t-test.

by wind speed, temperature, humidity, precipitations, solar radiation, weekend/working days and sensors elevation upon the sea level (fixed effects). A single random effect grouping factor was included, resuming the information about sensor ID and year, month, day, hour when the measure was performed (5,452 levels or intercept values).

Results are reported in Table 3 and show statistically significant reductions in terms of $PM_{2.5}$ and $PM_{10}$ values during the lockdown period in 2020 compared to the same period in 2019 accounting for potential confounders (PM2.5: Year (2020) coefficient = - 2.58 μg/m$^3$, 95% CI = -3.25 to -1.91 μg/m$^3$, $p < 0.0001$; PM10: Year (2020) coefficient = - 2.25 μg/m$^3$, 95% CI = -2.87 to -1.64 μg/m$^3$, $p < 0.0001$).

**Variation in terms of pollutants concentration between 2019 and 2020 by daily hours.** The presence of differences in terms of PM variations between the considered periods in 2019 and 2020 has been then inspected by daily hour intervals by including the year- hour interaction and the same set of confounders previously described in the regression. Adjusted means estimated by multivariate linear mixed effect models showed that both $PM_{2.5}$ and $PM_{10}$ variations were characterized by different trends as function of the daily hours interval considered (Fig 5 and S10 Fig).

**Table 3. Multivariate linear mixed effects model regression: Variation in terms of PM2.5 and PM10 between 2019 and 2020 accounting for confounders.**

| | PM2.5 | | PM10 | |
|---|---|---|---|---|
| Variable | Estimate (95% CI) | p | Estimate (95% CI) | p |
| *Year (2020)* | -2.58 (-3.25:-1.91) | *<0.0001* | -2.25 (-2.87:-1.64) | *<0.0001* |
| Working day (yes) | 11.15 (10.36:11.94) | <0.0001 | 10.15 (9.42:10.87) | <0.0001 |
| Average wind speed (m/s) | -4.33 (-4.99:-3.66) | <0.0001 | -3.96 (-4.57:-3.36) | <0.0001 |
| Temperature (˚C) | 1.59 (1.48:1.7) | <0.0001 | 1.51 (1.4:1.61) | <0.0001 |
| Sensors' elevation upon the sea level (m) | 0.02 (-0.04:0.08) | 0.4850 | 0.03 (-0.03:0.08) | 0.3002 |
| Humidity (%) | 0.54 (0.52:0.57) | <0.0001 | 0.49 (0.47:0.52) | <0.0001 |
| Precipitations (mm) | -9.55 (-11.2:-7.9) | <0.0001 | -8.85 (-10.36:-7.34) | <0.0001 |
| Solar radiation (W/m$^2$) | 0 (-0.01:0) | 0.0015 | 0 (-0.01:0) | 0.0001 |

Estimate (95% CI) = regression coefficient and 95% CI; p = p-value. The regression coefficient corresponding to "year (2020)" term quantifies the variation in terms of $PM_{2.5}$ and $PM_{10}$ pollutants concentration (μg/m$^3$) between 2019 and 2020 accounting for the other variables included in the model reported in the table.

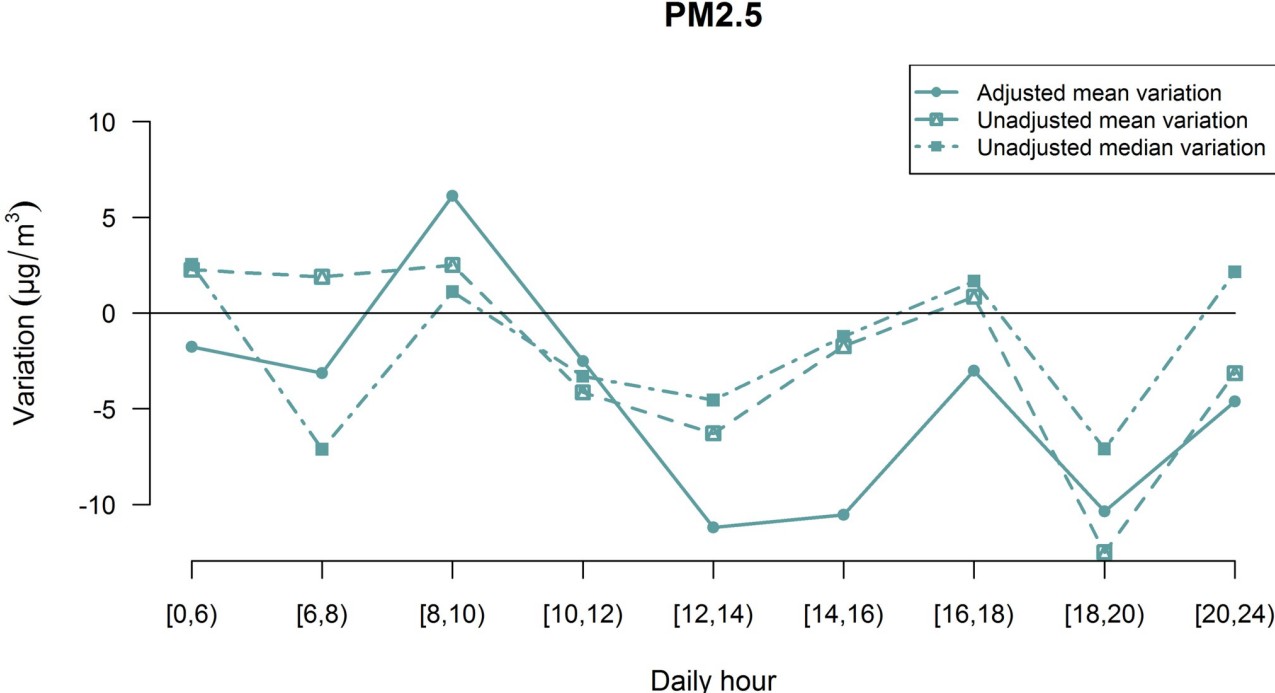

**Fig 5. PM$_{2.5}$ variations by daily hour interval.** Data are presented as adjusted mean variations, unadjusted mean variations and unadjusted median variations.

By inspecting the adjusted mean variations it is possible to observe a general reduction in terms of PM$_{2.5}$ and PM$_{10}$ between 2019 and 2020 during all day except from 8 to 10 in the morning, when a pollutants increase was estimated. Unadjusted mean and median variations showed conflicting trends of variations: discrepancies with respect to the adjusted mean variations could be partially explained by the non-gaussian distribution of PM measures variations during the different daily hours and by the lack of correction by confounders.

The adjusted variations estimated by linear mixed effect models in the whole sample and by sensor and daily hour (by performing sensor-level analyses) are graphically represented in Fig 6 and reported in S2 and S3 Tables. Results show strong consistency in terms of adjusted mean PM variations by daily hour intervals across sensors. A general pollutants increase was estimated from hour 8 to 10 in the morning for most of the sensors. In some cases increases were observed also from 10 and 12 and—sporadically—during evening/night, morning and afternoon hours.

An alternative approach consisting of a modification of the method described by Venter et al. [21] has been also evaluated to assess the impact of the COVID-19 related lockdown on pollutants concentration. According to this method, positive absolute changes in terms of pollutant concentration (observed—expected) indicate an increase in terms of PM compared to the expected values while negative changes a decrease in terms of pollutants concentration compared to the expected values. Positive and negative changes can be attributed to the COVID-19 related lockdown.

Results deriving from this approach were generally concordant with respect to the main findings reported in this section (S4 Table, S11 Fig). A negative change in terms of both PMs was estimated for all sensors during all daily hour intervals except for hours 8 to 10 in the morning, being characterized by a positive change for most sensors for both pollutants. PM$_{10}$

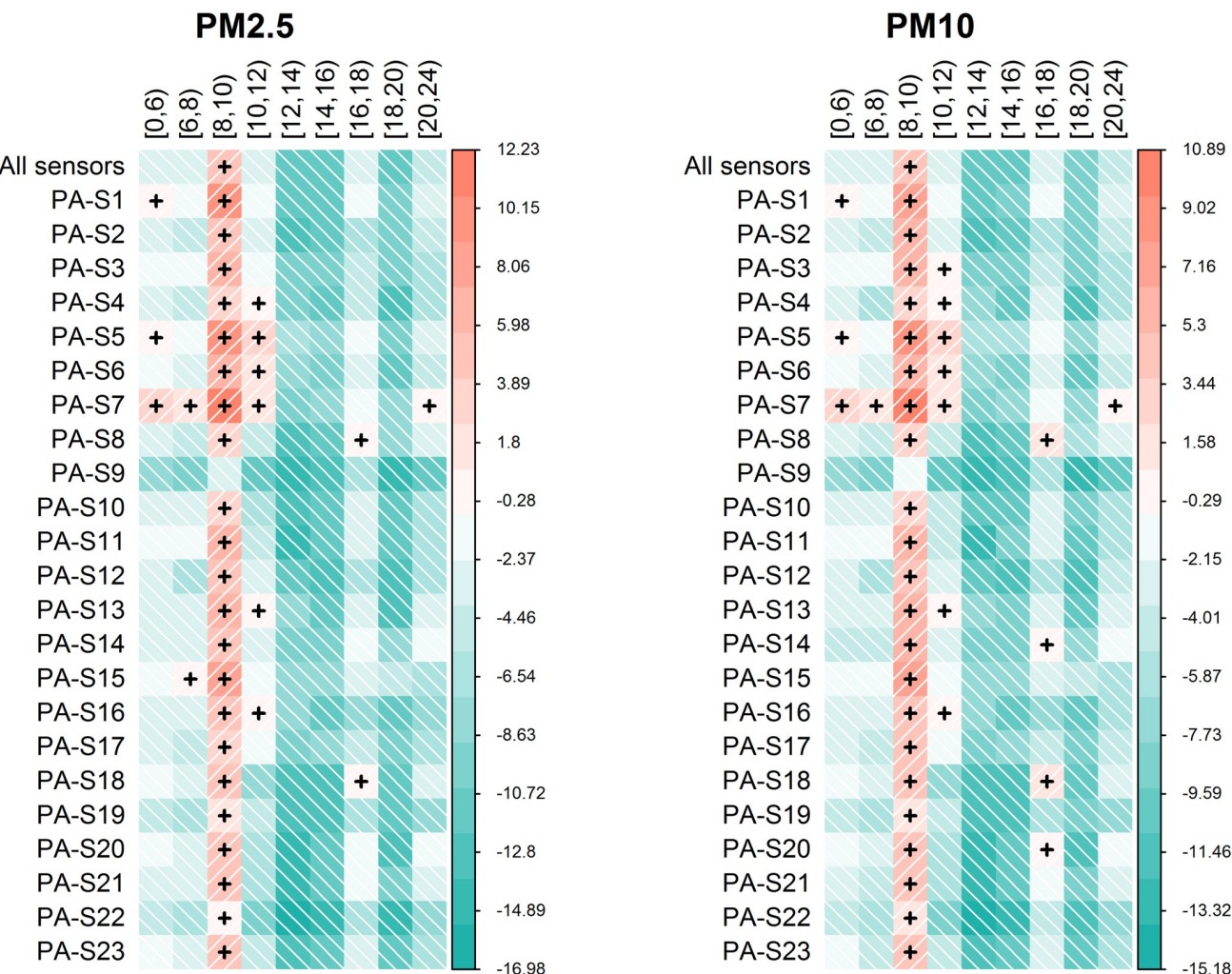

**Fig 6. Heatmaps showing adjusted mean variations in terms of PM$_{2.5}$ between the 2019 and 2020 study periods by sensor and daily hour intervals.** Each heatmap row represents graphically the adjusted mean variation in terms of PM$_{2.5}$ and PM$_{10}$ between 2019 and 2020 over the complete set of Purple Air sensors (All sensors) and by sensor (PA-S1 –PA-S23) over daily hours intervals defined. Shades of green are proportional to the estimated magnitude of the reduction in terms of PM$_{2.5}$ between 2019 and 2020; shades of red are proportional to the estimated magnitude of the increase in terms of PM$_{2.5}$ between year 2019 and 2020 as showed by the colors legend on the right side of each plot. The "+" symbol denotes an increase in terms of pollutants concentration between year 2019 and 2020.

showed positive changes for some sensors also during hours 10–12 while some sensors (PA-S7, PA-S8, PA-S13 and PA-S18) were characterized by a positive change also during afternoon/evening hours.

## Discussion

With the COVID-19 pandemic, which initially had its European epicenter in northern Italy at the end of winter 2020, a strict lockdown was imposed throughout Italy, as it happened previously in some regions of China and subsequently in the rest of Europe and America, as the COVID-19 cases spread and rose. For several weeks, most of the population was confined at home and all nonessential productive activities were closed. With these measures, a large part of the pollution sources was temporarily stopped, creating the ideal ground to perform

scientific studies to verify whether the limitation of traffic and shutdown of some activities had an impact on global and local air pollution. A large number of studies aiming at analyzing these aspects were carried out throughout the world, with results varying depending on the geographical areas covered. Some reports showed a global reduction in several pollutants worldwide [21], while other local-level studies obtained different results. Wang et al. [22] observed a significant reduction in all pollutants in 325 cities in China, especially in the northern cities, whereas Donzelli et al. [23] performed a similar analysis over a 7 months period for 3 cities in Tuscany, Italy, and reported different results depending on the pollutant ($NO_2$ levels changed significantly while PMs did not). A general study performed over western Europe performed by Menut et al. [24] taking into account also meteorological variables, showed a general reduction in terms of $NO_2$ throughout the continent, while the reduction in terms of PMs and $O_3$ was limited on a general level, but with pronounced local variations. Unlike other areas of the world, a severe reduction of all the pollutants during the lockdown in the Po Valley was not observed [9], indicating that traffic may not have a huge impact on pollution and that meteorological conditions probably play a role even more important than what was usually thought.

In this study, we analyzed the case of Pavia, a city in an area with poor air quality, where a dense network of sensors had been deployed a few months before as part of the European project PULSE, that had the aim to study air quality, among other things, in order to create a collaborative preventive system to help citizens and public health operators to reduce diseases risks and improve quality of life. There are several advantages deriving from the availability of a dense sensors network, such as the possibility to monitor local conditions, estimate pollution levels in each neighborhood and to analyze temporal trends. The choice of the time period considered in the study (i.e., between February and April 2019 and 2020) was motivated by the fact that unique conditions were applied in that period in 2020, as the total lockdown that was instituted had no precedents and was never repeated in the following months. In fact, even if restrictions were kept throughout 2020, the same level of shutdown of all nonessential activities and limitation to all movements were never reached again.

The results of our study show mainly two aspects: first, as expected and already demonstrated in literature, the influence of climatic factors on PM levels is extremely high, second, considering this difference, there has been a mildly significant reduction in PM levels in 2020 during the lockdown, even though pollution levels remained relatively high even without traffic and most productive activities. The latter aspect suggests that most of the relevant pollution sources in this particular geographical area are represented by the heating systems in private houses and commercial traffic, and, even with only essential services active, the peculiar climatic conditions characterized by long periods without wind are able to create dangerous levels of pollution. One interesting result is that looking at the official ARPA data alone, there is not a clear decrease in the PM levels in 2020 if all meteorological variables are not taken into account, whereas there is a definite decrease considering other pollutants. This appears to be in line with other studies conducted over Italy such as [23] and [24]. Considering the variety of pollution sources and local environmental and meteorological situations, the reasons for this are not easy to explain, and a precise characterization of the main pollutants' sources is needed to understand these dynamics. For example, $NO_2$ is usually produced from anthropogenic emissions such as industrial burning of fossil fuels like coal, oil and gas and vehicle exhaust [25]. A majority of the Particulate Matter, on the other hand, is formed from secondary formation, as PM is a generic measure that includes lots of different types of dusts, including soil residuals, sea salt, car pneumatics debris and all types of combustion processes residuals [26], including those used in old house heating systems. Traffic, which is the main activity that was reduced during the lockdowns, contributes less to PMs than it does to other pollutants such as

$NO_2$. This could explain why PM did not have the same reduction other pollutants had. Part of the PM produced in the area of Pavia could come from the agriculture sector, as intensive agriculture is diffuse in all the surrounding areas of Pavia, and was not limited during the lockdown, being food production an essential service. House heating systems, especially older ones, could also be responsible to the limitation in decrease of PM levels, as during the lockdown, with people spending most of their time at home. It is likely that house heating was more needed than during the previous year, considering that the average temperature in 2020 was lower than the one in 2019. Increased house heating could partially explain also the change in the daily patterns that was observed during the lockdown, with a high peak in the first hours of the morning. This result in fact suggests a different behavior of different emission sources, primarily house heating, as there is a pollution increase in the coldest hours of the day, and people staying at home in 2020, together with a lower outside temperature compared to the previous year, probably led to a higher use of heating. On the other hand, during 2019, without lockdowns, the same morning hours were the so-called "rush hours", characterized by increased vehicular traffic, that probably leads to an increase in other pollutants and a more constant increase in PMs that reaches higher levels in the central hours of the day.

PM levels could be influenced also by other climatic factors that are difficult to monitor, for example it has been noticed that eastern winds sometimes can bring sea salt from the Adriatic Sea, or southern winds, even if absent on the ground but strong on high altitudes, can bring sand from the Sahara Desert, that is deposited on the ground by rain or snow and remains there even after the rain dries completely. These kinds of phenomena could be partly responsible for the absence of a significant decrease of pollution levels during the lockdown if meteorological factors are not taken into account, for example during the month of March 2020 there has been occasional eastern winds that brought some sea salt that could have increased PM levels [10].

Unfortunately, identification of all the sources of pollution and their relative contribution to the local measurements is different due to the complexity in sources and processes. Nevertheless, pollution remains a serious problem in this region. The National Health Ministry of Italy estimated that living in the Po valley leads to a significant reduction of life expectancy [27], therefore wide interventions are needed to mitigate health risk related to air quality in northern Italy. Traffic limitations and introduction of new propulsion technologies, such as hybrid or electric engines, certainly contribute to an improvement of air quality, but are not enough to reduce health risk to safe levels, as even stopping nonessential traffic and closing all nonessential productive activities, the reduction in air pollution is mild.

## Supporting information

**S1 Fig. PM10 correlation matrix.** Correlation matrix (Pearson's correlation) of the PM10 measurements performed by seven Purple Air sensors co-located on a selected day.
(TIF)

**S2 Fig. PM2.5 correlation matrix.** Correlation matrix (Pearson's correlation) of the PM2.5 measurements performed by seven Purple Air sensors co-located on a selected day.
(TIF)

**S3 Fig. Range of the PM10 values measured by the seven co-located sensors.**
(TIF)

**S4 Fig. Range of the PM2.5 values measured by the seven co-located sensors.**
(TIF)

**S5 Fig. PM2.5 and wind variations in time in the considered periods in 2019 and 2020.** The upper plot shows 2019 data, the lower plot shows 2020 data.
(TIF)

**S6 Fig. PM2.5 concentration and temperature in the two considered periods in 2019 (upper plot) and 2020 (lower plot).**
(TIF)

**S7 Fig. PM10 and temperature variations in time in the considered periods in 2019 and 2020.** The upper plot shows 2019 data, the lower plot shows 2020 data.
(TIF)

**S8 Fig. Visual representation of the cross-validation results for PM2.5 and PM10.** The x-axis represents the complexity parameter corresponding to different tree sizes while the y-axis represents the cross validation relative error.
(TIF)

**S9 Fig. Correlation between variations in terms PM2.5 and PM10 between year 2019 and 2020.**
(TIF)

**S10 Fig. PM10 variations by daily hour interval.** Data are presented as adjusted mean variations, unadjusted mean variations and unadjusted median variations.
(TIF)

**S11 Fig. Heatmaps showing the median difference between observed and LMM—predicted PM2.5 and PM10 during 2020 by sensor and daily hour intervals.** Each heatmap row graphically represents the median absolute differences between observed and predicted pollutants measurements by daily hours intervals. Shades of green indicate negative absolute differences between observed and predicted pollutants concentration during 2020, shades of red indicate positive absolute differences between observed and predicted pollutants concentration during 2020 as showed by the colour code legend on the right side of each plot. The "+" symbol denotes a positive absolute differences between observed and predicted pollutants concentration.
(TIF)

**S1 Table. Number of paired measurements available by sensor and hours.**
(DOCX)

**S2 Table. Adjusted mean variations in terms of PM2.5 between 2019 and 2020 by sensor and daily hours.** Each PurpleAir (PA) ID corresponds to a different sensor. In green: reduction in terms of PM2.5 between 2019 and 2020; in red: increase in terms of PM2.5 between 2019 and 2020.
(DOCX)

**S3 Table. Adjusted median variations in terms of PM10 between 2019 and 2020 by sensor and daily hours.** Each Purple Air (PA) ID corresponds to a different sensor. In green: reduction in terms of PM10 between 2019 and 2020; in red: increase in terms of PM10 between 2019 and 2020.
(DOCX)

**S4 Table. Root mean square error, mean absolute error and Pearson correlation coefficient r from the LMM regression method.**
(DOCX)

**S1 File. List of supporting images and tables cited throughout the manuscript.**
(DOCX)

## Author Contributions

**Conceptualization:** Riccardo Bellazzi.

**Data curation:** Daniele Pala, Vittorio Casella, Cristiana Larizza, Alberto Malovini.

**Formal analysis:** Daniele Pala, Alberto Malovini.

**Funding acquisition:** Vittorio Casella, Riccardo Bellazzi.

**Investigation:** Vittorio Casella, Cristiana Larizza, Alberto Malovini.

**Methodology:** Alberto Malovini, Riccardo Bellazzi.

**Software:** Alberto Malovini.

**Supervision:** Daniele Pala, Cristiana Larizza, Riccardo Bellazzi.

**Validation:** Vittorio Casella, Cristiana Larizza, Alberto Malovini.

**Writing – original draft:** Daniele Pala, Alberto Malovini.

**Writing – review & editing:** Daniele Pala, Alberto Malovini, Riccardo Bellazzi.

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
