## [Editor Report · Decision Letter 0]

22 Jun 2021

PONE-D-21-17702

Impact of COVID-19 Lockdown on PM Concentrations in an Italian Northern City: a year-by-year Assessment

PLOS ONE

Dear Dr Pala,

Thank you for submitting your manuscript to PLOS ONE. After careful consideration, we feel that it has merit but does not fully meet PLOS ONE’s publication criteria as it currently stands. Therefore, we invite you to submit a revised version of the manuscript that addresses the points raised during the review process.

See below comments.

We look forward to receiving your revised manuscript.

Kind regards,

Zongbo Shi

Academic Editor

PLOS ONE

Journal Requirements:

Additional Editor Comments (if provided):

There is considerable interests in understanding the impact of COVID-lockdown on air quality. There are a very large number of publications on this topic in the past year. Most studies used ground or satellite observations or both but few used low cost sensors. This research is very interesting and the data potentially highly valuable. The outcome has the potential for pollution control so it also has the potential to impact policy.

Many have compared air quality ground observation or satellite data before and after the lockdown or in 2020 with previous years. More recent studies pointed out the major problems of such direct comparison. Air pollution levels are dependent on 1) emissions, 2) meteorology, and 3) chemistry. The meteorology effect could include local scale parameters such as wind speed, wind direction, temperature, relative humidity, cloud cover, precipitation etc., but synoptic scale meteorology can also have a major impact, e.g., via long-range transport of pollutants. This background information is important for consideration.

The data are interesting and probably of high quality. Some of the results, such as little change in PM levels, changes in diurnal patterns are all interesting. The manuscript will require substantial additional work, including more targeted data analyses in order to account for the confounding factors, as the authors pointed out. Only when accurate regression models or machine learning based models (such as random forest models) are established to predict the levels of PM in 2019 with the meteorological conditions in 2020 (or vice versa), you won’t be able to compare the two years.

For this specific manuscript, there are some issues that the authors should address or clarify.

1) Detailed information on the sensor calibration: how each of the sensors are calibrated? Are all the sensors been co-located at one of the ARPA stations? Does the performance of the sensor change with time, and how this is controlled? How do the sensors compare with each other?

2) Time series figures or raw data for each sensor should be provided, perhaps as a supplementary table or dataset. It may be easier to separate the sensor location into urban background, traffic and rural background (if any) and present the average data. If so, the original raw data should also be provided.

3) What data are available from the ARPA stations and how they are measured? It mentioned ARPA sensors – what are they?

4) Figure 1 – short-term air pollution levels are highly dependent on meteorology. Is it really meaningful to compare the concentration of PM10 levels in late Feb to March in 2020 with that in 2019? What is interesting is that there are large differences for the two stations (2020 vs 2019). It would be important to explain why this is the case. The quality of the figure is not so good but this can be rectified at a later stage. You may want to put the two figures into one.

5) Line 220 – this should be in the methodology?

6) Line 226 – this section explored the effect of meteorology on air pollution levels. There are a number of figures but the key messages could be made clearer. Why only wind speed and temperature? Other meteorological conditions such as wind direction, relative humidity, cloud cover, precipitation, and back trajectories all could contribute to the variations in air quality. The key point of this section is to establish the regression models so that you can predict the concentrations under same meteorological conditions whether that is in 2020 or 2019. Although this regression model is not as good as machine learning based techniques, such as based on random forest algorithms (you should be able to find relevant papers), it has been used by Zenter et al. (https://www.pnas.org/content/pnas/117/32/18984.full.pdf). Only by doing this, you can compare the data in 2019 with that in 2020.

7) Table 2: do not understand this. For example, what does intercept mean?

8) Section starting line 308: this is interesting. Consider putting data in Table 3 and 4 in the SI and present the data in figures. It would be much easier to see the trend.

9) Is there longer-term changes in PM levels in the study region? For example, are the PM levels reducing in the previous years? If so, then the PM levels will be lower in 2020 whether or not there is a lockdown. You may need to take this “trend” into account as well. This is called “detrend”.

10) Discussions – this seems very general. It would be important to explain your results, e.g., why there is no obvious change in PM levels? We know the emission reduced in 2020 as a result of the lockdown. Is it due to the sensor uncertainty, or meteorology difference in 2019 and 2020, or is it due to the negligible contribution to PM levels from road traffic? Or is it due to changing chemistry? Do you see variations in different types of sites, for example, do you see changes in PM levels at roadside sites but not at the urban background sites? The changing diurnal patterns are really interesting and suggest different emission sources. Are there any other data, such as NO2, CO, BC data from the monitoring stations that can potentially help to interpret the results better? The second part of the discussion should focus on the implications of the results for air pollution control in the study region.
---

## [Author Response · Author response to Decision Letter 0]

6 Aug 2021

We would like to thank the Reviewer for the positive comments on our study and for raising interesting points regarding our research. The comments have been very beneficial to improve the quality of the paper and better conveying its message. It is true that there is considerable interest on the effect of the lockdown on air pollution, as the unique situation of last year caused an unprecedented opportunity to explore the real impact of traffic and some productive activities on air quality. The phenomenon is really complex, as it is influenced by a large number of external factors. 

In this revision process, we addressed all the points of the review, adding new analyses and editing the manuscript where required. Concerning the list of comments, the detailed modifications are the following:

1) Detailed information on the sensor calibration: how each of the sensors are calibrated? Are all the sensors been co-located at one of the ARPA stations? Does the performance of the sensor change with time, and how this is controlled? How do the sensors compare with each other?

Purple Air sensors are already calibrated during their production before being made available for purchase, as stated in the official documentation. To increase the precision of their measurements, we applied a simple recalibration method, explained in lines 176-204 of the manuscript, where we used the data coming from the official ARPA sensors as reference, since those sensors are calibrated with reliable procedures according to the European standards (see point 3). A parallel study that was performed at the University of Pavia showed that co-located Purple Air sensors measure PMs with high correlation, and therefore that the sensors are expected to provide consistent results between each other. For this reason we used only one sensor colocated with the ARPA one to estimate the correction to be made, and then applied it to all the other sensors. Considering that Pavia is a small city with no significant climatic differences within different zones, we assume that a possible change in performance over time does not affect some sensors more than others. The calibration process is repeated every year. We added a more detailed explanation in lines 194-203 and reported the results of the co-location studies in supplementary figures 1-4.

2) Time series figures or raw data for each sensor should be provided, perhaps as a supplementary table or dataset. It may be easier to separate the sensor location into urban background, traffic and rural background (if any) and present the average data. If so, the original raw data should also be provided.

The raw sensor data will be provided on a GitHub folder, both for the Purple Air sensors and the ARPA ones. Pavia is a small city and does not have significant changes in urban landscape, as there are not extremely urbanized areas and industrial sites inside the city borders. We also did not install sensors in rural areas (which are outside of the municipality borders). Nevertheless, the central area of the city is mostly closed to traffic, so we divided the sensors according to their locations into two categories: center and traffic areas. These categories have been added to the raw data. Most sensors belong to the traffic areas category and significant differences in the data from the two categories were not observed.

3) What data are available from the ARPA stations and how they are measured? It mentioned ARPA sensors – what are they?

ARPA Lombardia (Environmental Protection Regional Agency of the region of Lombardy) is a public agency that has the aim of measuring and treating environmental data of the region the city of Pavia is located in. Through numerous sensors scattered throughout the region, ARPA collects a large quantity of data about air pollution, meteorology, agriculture, sole status etc.

In Pavia, there are two official ARPA air quality monitoring stations, they are high-quality fixed stations that measure several pollutants (NOX, SO2, CO, O3, PM10 and PM2.5) at regular time intervals. These sensors are calibrated continuously using other commercially available instruments that can be used as reference according to Italian or European laws (https://www.arpalombardia.it/Pages/Aria/Rete-di-rilevamento/Qualit%C3%A0-dei-dati/Taratura-degli-strument.aspx?firstlevel=Rete%20di%20rilevamento) and therefore can be considered very reliable. Weather parameters are collected as well, and data is freely available upon request on a dedicated portal.

We added this information in the manuscript, lines 132-149. 

4) Figure 1 – short-term air pollution levels are highly dependent on meteorology. Is it really meaningful to compare the concentration of PM10 levels in late Feb to March in 2020 with that in 2019? What is interesting is that there are large differences for the two stations (2020 vs 2019). It would be important to explain why this is the case. The quality of the figure is not so good but this can be rectified at a later stage. You may want to put the two figures into one.

It is true that pollution levels are highly dependent on meteorology, and unfortunately spring months like March are the most meteorologically unstable months in which wind, humidity, pressure etc. can vary very quickly and influence air pollution’s measurements. The differences between 2019 and 2020 are due to this variability, as 2019 had a warmer spring mostly dominated by high pressure systems and higher temperatures, whereas 2020 had a generally lower temperature and higher instability. The aim of this image is to show the effects of this variability and prove that meteorology can affect the measurements in a significant way. Since the lockdown was established in this period, comparing data in this unstable month was not a choice, but our analyses explicitly consider the effects of meteorological variables to avoid confounding factors (see also point 6). Comments about this have been added in lines 268-271.

5) Line 220 – this should be in the methodology?

We thank the Reviewer for the suggestion and moved it to the Methods section (lines 203-208).

6) Line 226 – this section explored the effect of meteorology on air pollution levels. There are a number of figures but the key messages could be made clearer. Why only wind speed and temperature? Other meteorological conditions such as wind direction, relative humidity, cloud cover, precipitation, and back trajectories all could contribute to the variations in air quality. The key point of this section is to establish the regression models so that you can predict the concentrations under same meteorological conditions whether that is in 2020 or 2019. Although this regression model is not as good as machine learning based techniques, such as based on random forest algorithms (you should be able to find relevant papers), it has been used by Zenter et al. (https://www.pnas.org/content/pnas/117/32/18984.full.pdf). Only by doing this, you can compare the data in 2019 with that in 2020.

We thank the Reviewer for highlighting these aspects: we updated the analyses by including additional potential confounding factors in the models and tested an additional methodology to estimate the impact of the lock-down on pollutants concentration.

The set of confounders considered in the analyses now include: wind (m/s), temperature (°C), humidity (%), precipitations (mm) and solar radiation (W/m2). The correlation between the updated set of variables and PM2.5 and PM10 concentrations has been extensively explored as described in the Results section with title “Effect of potential confounders on pollutants concentration”. In particular, an updated approach to data filtering has been applied as reported in the following. Based on the scatterplots in Fig 6 we observed a non-linear relationship between wind speed and pollutants concentration (Fig 4A and Fig 4F). With the aim to identify informative wind speed cut off values able to distinguish subpopulations of measurements, univariate regression trees were fitted including wind speed as predictor while PM2.5 and PM10 as dependent variables in turn. By imposing a single split to the regression tree algorithm, a wind speed of 2.15 m/s was identified as the most informative threshold to stratify both PM2.5 and PM10 levels. Further, plots in Fig 4C and Fig 4H evidenced that pollutants concentration did not vary with respect to humidity when humidity values were below ~ 20%, highlighting a potential bias in terms of measurements accuracy when the confounder value is below this threshold. It was then decided to focus on measures performed when the wind speed was below 2.15 m/s and humidity > 20% to avoid confounding effects.

As an additional approach to estimate the effect of the lock-down on pollutants concentration, a methodology based on the method described in Venter et al. (https://www.pnas.org/content/pnas/117/32/18984.full.pdf) has been applied as described in the Supplementary Data section. In details, the implemented methodology consists of the following steps:

a) Train linear mixed model regression models (LMM) using data from 2019 at a sensor level, using PM2.5 and PM10 levels in turn as dependent variable and the following predictors as independent variables: working day, wind, temperature, humidity, precipitations, solar radiation, day/month, daily hour categories. The day/month information is used as random effect grouping variable, while the remaining ones as fixed terms.

b) Apply the LMM model trained on data from 2019 to forecast PM2.5 and PM10 concentrations during 2020.

c) Using the predicted pollutants concentration as benchmark, compare the predicted and the observed PM2.5 and PM10 values to estimate the absolute change (observed – predicted) in terms of pollutants concentration. Positive changes indicate an increase in terms of pollutants concentration compared to the expected values, negative changes a decrease in terms of pollutants concentration compared to the expected values.

The Pearson correlation coefficient r between observed and predicted PM2.5 and PM10 concentrations during 2020 was + 0.51 for PM2.5 and + 0.52 for PM10 (Supplementary Table 4). The median value of the absolute differences between observed and predicted pollutants concentration by sensor and daily hour showed trends concordant with what estimated by the method used in our manuscript (Supplementary Figure 8 vs. Figure 9).

7) Table 2: do not understand this. For example, what does intercept mean?

Table 2 reports the regression coefficients, 95% confidence intervals and p-values corresponding to the set of variables included in the multivariate linear mixed effects model regression fitted to estimate the mean variation in terms of PM2.5 and PM10 between 2019 and 2020 accounting for confounders. The regression coefficient corresponding to the “year (2020)” term quantifies the average variation in terms of PM2.5 and PM10 pollutants concentration between 2019 and 2020 accounting for potential confounding effect of the other variables included in the model and reported in the table. For sake of clarity the coefficients and significance corresponding to the intercept term have been now removed from Table 2. These aspects have been described more in the detail in Table 2 legend and in the corresponding text section.

8) Section starting line 308: this is interesting. Consider putting data in Table 3 and 4 in the SI and present the data in figures. It would be much easier to see the trend.

We made the proposed change, and we thank the Reviewer for the suggestion. Table 3 and Table 4 have been moved to Supplementary Data section (Supplementary Table 2 and Supplementary Table 3) and replaced by the two Heatmaps in Figure 9 graphically resuming the same information.

9) Is there longer-term changes in PM levels in the study region? For example, are the PM levels reducing in the previous years? If so, then the PM levels will be lower in 2020 whether or not there is a lockdown. You may need to take this “trend” into account as well. This is called “detrend”.

There is some evidence of a reduction trend in all the area in the last years ( https://www.infodata.ilsole24ore.com/2021/01/31/qualita-dellaria-italia-migliorata-negli-ultimi-cinque-anni-cosa-misura-snpa/?refresh_ce=1 ), although with notable fluctuations. Looking at the data gathered by the Italian National Environmental Protection System, it can be noticed that the reduction trend appears less evident after 2018, with even a little increase, probably not significant, in the PM10 concentrations in 2020. The article itself states that the meteorological variability could have played an important role in the measurements’ variations, as in 2019 and 2020 temperatures were generally higher and precipitations lower than the previous years. Therefore, we do not assume that the general trend of the last years could influence the difference in PM2.5 and PM10 concentrations between 2019 and 2020. We added a few lines about this (lines 210-224) with the proper citations.

10) Discussions – this seems very general. It would be important to explain your results, e.g., why there is no obvious change in PM levels? We know the emission reduced in 2020 as a result of the lockdown. Is it due to the sensor uncertainty, or meteorology difference in 2019 and 2020, or is it due to the negligible contribution to PM levels from road traffic? Or is it due to changing chemistry? Do you see variations in different types of sites, for example, do you see changes in PM levels at roadside sites but not at the urban background sites? The changing diurnal patterns are really interesting and suggest different emission sources. Are there any other data, such as NO2, CO, BC data from the monitoring stations that can potentially help to interpret the results better? The second part of the discussion should focus on the implications of the results for air pollution control in the study region.

The discussion has been extended according to the new results and the Reviewer’s comments, especially from line 480.

---

## [Decision Letter · Decision Letter 1]

22 Nov 2021

PONE-D-21-17702R1Impact of COVID-19 Lockdown on PM Concentrations in an Italian Northern City: a year-by-year AssessmentPLOS ONE

Dear authors,

Thank you for submitting your manuscript to PLOS ONE. After careful consideration, we feel that it has merit but does not fully meet PLOS ONE’s publication criteria as it currently stands. Therefore, we invite you to submit a revised version of the manuscript that addresses the points raised during the review process.

Please see detailed comments below. And also refer to the annotated manuscript. 

We look forward to receiving your revised manuscript.

Kind regards,

Zongbo Shi

Academic Editor

PLOS ONE

Journal Requirements:

Additional Editor Comments:

Thanks for revising the manuscript. The new models constructed enabled the identification of the difference in the two years. This is a good improvement.

There are major presentational issues. The overall English presentation is very poor with numerous English errors.

There are two many figures and some could be moved to the SI.

The key messsages are: by comparing the raw data, there are no detectable difference during the periods; by adjusting for meteorological conditions, a difference can be identified. And your results should focus on your message rather than presenting a large number of figures, some of which are not particulalry relevant to your key massage.

We know that meteorological conditions affect PM levels - this is not new so do not spend too much time on this issue. You need to identify the difference due to lockdown effects.

The Discussions are very poorly written. A majority of such discussions should be in the Introduction. Discussions should include:

1) Why you have seen a reduction in 2020: you should look at literature data and emission inventory to see what are the main sources of particles; you can then explain your data by linking to mobility changes - which leads to lower levels of PM / NOx emissions.

2) Why you did not see a reduction in 2020 in raw data - explain which meteorological factor(s) contributed to this. Basically you results showed that meteorological conditions in 2020 was not as good as in 2019, so the observed concentrations are higher than those if under 2019 meteorological conditions.

3) What are the implications of your results: something like this; Such a lockdown did not change PM levels much - so more efforts should be focused on sources other than traffic.

Reviewers' comments:

Reviewer's Responses to Questions

**Comments to the Author**

1. If the authors have adequately addressed your comments raised in a previous round of review and you feel that this manuscript is now acceptable for publication, you may indicate that here to bypass the “Comments to the Author” section, enter your conflict of interest statement in the “Confidential to Editor” section, and submit your "Accept" recommendation.

Reviewer #1: All comments have been addressed

2. Is the manuscript technically sound, and do the data support the conclusions?

Reviewer #1: Yes

3. Has the statistical analysis been performed appropriately and rigorously? 

Reviewer #1: Yes

4. Have the authors made all data underlying the findings in their manuscript fully available?

Reviewer #1: Yes

5. Is the manuscript presented in an intelligible fashion and written in standard English?

Reviewer #1: Yes

6. Review Comments to the Author

Reviewer #1: The paper reports the impact of the COVID-19 lockdown on the urban PM pollution in an Italian city. The authors did not found a drastic decrease in PM pollution in the Po Valley during the lockdown, which is different from other areas worldwide. The paper is an interesting case study that provides information for understanding factors influencing air pollution in Italy. I can see that the authors have improved the paper significantly following Referee1’s comments. Therefore, I would suggest the paper to accepted after minor revisions.

My comments:

1) Line 70, I do not think that precipitation could disperse air pollutants.

2) The 10 and 2.5 in terms PM10 and PM2.5 should be subscript.

3) Line 126, I suggest the authors add one more sentence to describe the measurement principle of PurpleAir sensors. Is it based on laser measurement?

4) Figure 1, the word is too small to read. Captions of X-axis and Y-axis are missing.

5) Line 274 and Table 1, what is SO?

6) Figure 4, the word is too small to read.

7) Figures 5 and 6 could be combined as one figure.

8) The first paragraph in Discussion is unnecessary or should not be in Discussion section. Also, from my perspective, the discussion is weak. I would expect more discussion based on their own data and literatures.

7. PLOS authors have the option to publish the peer review history of their article (what does this mean?). If published, this will include your full peer review and any attached files.

Reviewer #1: No

---

## [Author Response · Author response to Decision Letter 1]

23 Dec 2021

We would like to thank the Reviewers for revising our manuscript and suggesting useful improvements. We tried to address each point editing the manuscript in the best possible way following the suggestions. Our specific modifications and comments are listed below:

Editor’s Comments:

There are major presentational issues. The overall English presentation is very poor with numerous English errors.

We double checked the manuscript again and corrected some mistakes.

There are two many figures and some could be moved to the SI.

We moved Figures 3,6 and 8 to the SI.

The Discussions are very poorly written. A majority of such discussions should be in the Introduction. Discussions should include:

1) Why you have seen a reduction in 2020: you should look at literature data and emission inventory to see what are the main sources of particles; you can then explain your data by linking to mobility changes - which leads to lower levels of PM / NOx emissions.

2) Why you did not see a reduction in 2020 in raw data - explain which meteorological factor(s) contributed to this. Basically you results showed that meteorological conditions in 2020 was not as good as in 2019, so the observed concentrations are higher than those if under 2019 meteorological conditions.

3) What are the implications of your results: something like this; Such a lockdown did not change PM levels much - so more efforts should be focused on sources other than traffic.

We thank the Reviewer from the suggestion, our discussion was in fact too much focused on a general discourse rather than an accurate dissertation on our findings and analysis. We modified the entire discussion section, deleting the first paragraph and going deeper on the comments concerning our analyses.

#Reviewer 1

We thank the Reviewer for the appreciation of our paper and the useful comments. We addressed all the minor issues as follows:

1) Line 70, I do not think that precipitation could disperse air pollutants.

That is true, precipitation does not disperse pollutants alone, although long lasting rain can push them to the ground. Anyway, we removed the sentence from the paper.

2) The 10 and 2.5 in terms PM10 and PM2.5 should be subscript.

We edited the text and made them subscripts.

3) Line 126, I suggest the authors add one more sentence to describe the measurement principle of PurpleAir sensors. Is it based on laser measurement?

We added a description of the measurement principle as suggested.

4) Figure 1, the word is too small to read. Captions of X-axis and Y-axis are missing.

We modified Figure 1 and increased the labels dimension.

5) Line 274 and Table 1, what is SO?

This was a typo, what we actually meant was NO. We thank the reviewer for making us notice it.

6) Figure 4, the word is too small to read.

We modified Figure 4 and increased the labels dimension.

7) Figures 5 and 6 could be combined as one figure.

In order not to decrease the dimension of the figures too much and make them difficult to analyze, we moved figure 6 in the SI.

8) The first paragraph in Discussion is unnecessary or should not be in Discussion section. Also, from my perspective, the discussion is weak. I would expect more discussion based on their own data and literatures.

We removed the first paragraph and, following also the other Reviewer’s suggestions, edited the discussion section adding more insights on our results.

---

## [Editor Report · Decision Letter 2]

17 Jan 2022

Impact of COVID-19 Lockdown on PM Concentrations in an Italian Northern City: a year-by-year Assessment

PONE-D-21-17702R2

Dear author,

We’re pleased to inform you that your manuscript has been judged scientifically suitable for publication and will be formally accepted for publication once it meets all outstanding technical requirements.In particular, the editor identified a series of editorial and English issues that will require close attention.

Kind regards,

Zongbo Shi

Academic Editor

PLOS ONE

Additional Editor Comments:

Line 24: replace “other” with “some”; as shown in previous studies, there are areas that do not show dramatic decrease and sometimes even increases.

Line 46: There is no evidence to suggest that it is “generated most likely in a market in Wuhan”. Please delete this type of statements that are not scientifically confirmed. It is widely accepted that “the virus is firstly reported in Wuhan, China”

Line 110: change “peculiar” to “particular”

Line 155: “Analysed data” – change to “Data analysed”

Line 189 to 195: use subscript for 2.5 and 10 for PM2.5 and PM10; also check throughout the paper. Line 261 and 265, 268for example.

Line 196: you can’t say that “it is enough to locate”; in reality, it is not a best practical. So you can only say that “we co-located one sensor close to the ARPA station…”

Line 255 – change “it is reasonable to assume” to “we assumed”

Line 271: this argument makes no sense: Met conditions will be almost the same at the two stations so the difference could not possibly be due to meteorology, particularly in such a small city. The fact that the two sensors are behaving so differently put serious doubts on the quality of the data. Are there any official observational stations nearby? I can’t see Figure 1 so can’t make a judgement. Did you show daily data in Fig 1? Are there missing data from one of the sites? In any case, this paragraph must be revised.

Throughout the paper, you mentioned year 2020 and year 2019. But you only mean the study periods so be more precise. Suggest to change to “during the study periods in 2020 or 2019”

Line 406. “… the sensors, in some cases…” change to “… the sensors. In some cases,…”

Line 464: “industrial sources” are likely to be important in the region and they could affect PM levels via long range transport. There should be lots of data showing this.

Line 472: “biomass burning, and electricity production” - I am not sure biomass burning is an important source of NO2. And electricity production usually use fossil fuels so these repeats the previous sentence. Suggest to delete these two terms.

Line 473: “NO2 is also the result of anthropogenic emissions such as combustion of coal and oil [26], 26], and both these pollutants are produced by vehicular traffic, especially NO2” : this repeats somewhat the previous sentence so may be deleted.

Line 474-477: “Particulate Matter, on the other hand, is a generic measure that includes lots of different types of dusts, including soil residuals, sea salt, car pneumatics debris and all types of combustion processes residuals [27], including those used in old house heating systems.” Suggests to change to “A majority of the particulate matter, on the other hand, is formed from secondary formation”

Line 477: “It is therefore possible that traffic, which was the major pollution source that was stopped

during the lockdown, contributes less to PMs than it does to other pollutants such as SO2 and NO2, and this could explain why PM did not have the same reduction other pollutants had.” Change to “Traffic, which is the main activity that was reduced during the lockdowns, contributes less to PMs than it does to other pollutants such as NO2. This could explain why PM did not have the same reduction other pollutants had”

Line 484: “…home, it is possible to assume that house heating was more needed than during the previous year, also considering that, according to our data, the average temperature in 2020 has been lower than the one in 2019” change to “…home. It is likely that house heating was more needed than during the previous year, considering that the average temperature in 2020 was lower than the one in 2019”

Line 486: “be responsible for” change to “contributed to”

Line 487: “Independently from house heating, a lower temperature can be responsible for an increased concentration of pollutants by itself, as cold air tends to be more dense and press pollutants towards the ground, especially if wind is absent and humidity is high or fog occurs” This should be deleted because you have accounted for the meteorological difference in your study. You can’t see any difference before accounting for this and you did after you accounted for the met conditions.

Line 504: “is really hard, as sources are numerous, nevertheless, pollution remains” change to “is different due to the complexity in sources and processes. Nevertheless, pollution remains…”

Line 508: “this situation” change to “air quality”

Line 509: “stopping traffic completely” change to “stopping non-essential traffic”

Line 510: “and can be noticed only through a thorough analysis that takes into consideration the influence of all meteorological and confounding factors”: delete this

---

## [Editor Report · Acceptance letter]

22 Feb 2022

PONE-D-21-17702R2 

Impact of COVID-19 Lockdown on PM Concentrations in an Italian Northern City: a year-by-year Assessment 

Dear Dr. Pala:

I'm pleased to inform you that your manuscript has been deemed suitable for publication in PLOS ONE. Congratulations! Your manuscript is now with our production department. 

Kind regards, 

on behalf of

Dr. Zongbo Shi 

Academic Editor

PLOS ONE